# Embodied Learning Through Immersive Virtual Reality: Theoretical Perspectives for Art and Design Education

**DOI:** 10.3390/bs15070917

**Published:** 2025-07-07

**Authors:** Albert L. Lehrman

**Affiliations:** Faculty of Education, Department of Art Education, Charles University, Magdalény Rettigové 4, 110 00 Prague, Czech Republic; treblasyd04@gmail.com or albertlehrman@gmail.com; Tel.: +420-777101241

**Keywords:** embodied cognition, immersive virtual reality, art education, design pedagogy, spatial reasoning, creative thinking, sensorimotor learning, educational technology

## Abstract

A significant development in pedagogical strategies which make use of the principles of embodied cognition can be found within the implementation of Immersive Virtual Reality (IVR) into art and design education. This theoretical study investigates how IVR-mediated embodiment enhances spatial thinking and creative problem-solving in art and design education by examining the taxonomy of embodied learning and principles of embodied cognition. The pedagogical affordances and limitations of IVR for creative learning are analyzed through a combination of empirical research and case studies, such as the Tangible and Embodied Spatial Cognition (TASC) system and Tilt Brush studies. Through gesture, spatial navigation, and environmental manipulation, IVR provides numerous possibilities for externalizing creative ideation; however, its implementation requires negotiating contradictions between virtual and physical materiality. IVR-based educational technologies have the potential to revolutionize teaching and learning. The goal of this paper is to provide educators with a theoretically grounded framework for applying embodied practices in IVR-based learning environments, while also acknowledging the current limitations of this technology.

## 1. Introduction

As strategies for teaching and learning continue to evolve within the 21st century landscape, Immersive Virtual Reality (IVR) marks a pivotal change in the way we use and perceive the affordances of new media and technology. The implementation of IVR into the modern classroom marks a significant shift, diverging from traditional tools towards more embodied and interactive approaches to teaching and learning. In keeping with the theory of embodied cognition, which proposes that interactions between one’s body and the outside world may facilitate cognitive processes ([44]; [37]), IVR presents novel opportunities for expressing creative ideas through gesture, spatial navigation, and environmental manipulation. Building upon current concepts of embodied cognition, such as [17]’s ([17]) taxonomy of embodied learning and [44]’s ([44]) six principles of embodied cognition, this investigation examines how IVR-based teaching and learning can impact creative collaboration, problem-solving, and spatial awareness in the field of art and design education.

Art and design educators have traditionally put great emphasis on studio-based learning, with manipulation of real, tangible materials being the primary means to foster spatial exploration and a deeper understanding of artistic principles, which are necessary to enhance students’ design thinking abilities. These embodied practices, inherent in art and design education, can now be expanded, improved, and transformed in previously unthinkable ways thanks to the development of IVR-based technologies. Through the creation of virtual environments that respond to natural body movements and gestures, IVR provides learners with unique opportunities to engage in and interact with design problems, potentially enhancing their understanding of spatial relationships, material properties, and design constraints.

In addition to these notable benefits of IVR-based technologies for art and design education, contemporary philosophers like [27] ([27]) suggest that IVR provides unique opportunities to investigate the fundamental questions of our conscious experiences and sense of “being”. As suggested by [27] ([27]), “VR is the best technological metaphor for the conscious experience we have,” implying that immersive virtual environments not only provide students with novel tools with which to investigate the nature of perception itself but also offer new means of artistic expression. This sentiment is supported by the phenomenological tradition, as noted in the works of [14] ([14]) and [26] ([26]), who emphasize that physical embodiment constitutes consciousness and cognitive experience ([37]).

Although the potential of IVR is seemingly limitless, its application poses further debates regarding virtual and physical materiality ([28]), leading to ethical concerns regarding perception mediated by immersive technology ([27]). This research attempts to give educators a theoretically based framework for utilizing embodied cognition in virtual learning environments by examining various case studies, serving as examples of IVR-based design pedagogy.

This paper aims to provide a theoretical contribution to the existing research by synthesizing the current knowledge on embodied cognition within IVR-based educational practices in order to offer a thorough framework for understanding their implementation within the field of art and design education. By examining established theories and identifying patterns amongst existing studies, this investigation will present an overview of the current pedagogical implications of embodied learning in virtual environments. Through the integration of insights from cognitive science, educational technology, and design pedagogy, this synthesis is intended to provide art and design educators with theoretically grounded perspectives for utilizing IVR’s unique affordances while acknowledging its current limitations, serving as a basis for future empirical investigations into the efficacy of embodied virtual learning in creative disciplines.

The primary research questions include the following:How do embodied cognition principles (e.g., Wilson’s six principles) manifest in IVR-based art/design pedagogy?What are the practical implications and limitations of IVR for fostering creativity and collaboration in design education?

## 2. Methodology: Theory-Driven Translational Synthesis

In order to explore how embodied cognition theory can guide the ethical and effective application of IVR in art and design education, this study uses a three-phase qualitative synthesis. Grounded in the notion that body–environment interactions shape our cognition ([44]), the study incorporates two major theoretical frameworks including: the embodiment taxonomy, put forward by [17] ([17]), which categorizes IVR experiences based on the level of physical interaction, and the six principles of embodied cognition proposed by [44] ([44]), which provide a framework for examining how sensorimotor interactions influence learning and creativity in IVR. Methodologically, this work adapts [39]’s ([39]) thematic synthesis, in its analysis, to prioritize embodied cognition metrics (such as haptic feedback and gestural congruency). The adopted three-phase design was developed as follows:

### 2.1. Phase 1: Selection and Synthesis of Interdisciplinary Studies

Peer-reviewed literature from 2010 to 2023 was thoroughly reviewed using Web of Science, Scopus, and specialized design education journals. Three conceptual clusters were combined in the search strategy: pedagogy (“art education” OR “design pedagogy” OR “materiality”), technology (“immersive virtual reality” OR “IVR” OR “haptics”), and embodiment (“embodied cognition” OR “sensorimotor learning”). After thorough screening of 1532 initial records, 89 studies fulfilled the inclusion criteria addressing RQ1’s emphasis on the manifestation of embodied principles in IVR pedagogy. The systematic review process followed the PRISMA guidelines as illustrated in Figure 1.

Quality assessment protocols ensured methodological rigor across all included studies. Each study was evaluated for theoretical alignment, research design validity, and relevance to embodied learning in IVR contexts. Studies lacking sufficient methodological rigor or unclear theoretical grounding were excluded from the synthesis. This systematic approach refined the initial 1532 records to 89 studies that met all inclusion criteria and quality thresholds.

Inclusion criteria: Theoretical works discussing the concepts of embodied cognition; empirical studies examining the use of IVR in creative education, design, or the arts; philosophical evaluations of embodiment within virtual environments; research on embodied creativity, metaphors, and interaction in artistic settings; studies on sensorimotor engagement and haptic feedback in virtual settings; critical analyses of constraints and ethical considerations of the use of IVR educational settings.

Priority for this synthesis was given to the following cases: [28]’s ([28]) identification of the immateriality paradox, theoretical frameworks of embodied learning, and empirical studies measuring the cognitive impacts of IVR, such as [5]’s ([5]) TASC system study. This methodical approach ensured correlation to a fundamental issue for IVR-based art and design education, aligning students’ engagement with artistic media to the spatial affordances of IVR.

### 2.2. Phase 2: Thematic Analysis Using Embodied Cognition Frameworks

Four complementary frameworks were used to structure the analysis, which employed iterative, theory-driven coding to determine how IVR-mediated environments facilitate learning and creative cognition. This multi-faceted approach allowed for a detailed evaluation of embodied interactions in art and design pedagogy.

#### 2.2.1. Frameworks of Embodied Cognition

I. [44]’s ([44]) six principles of embodied cognition serve as the primary lens, revealing the following:Environmental coupling corresponds to [44]’s ([44]) notions that “cognition is for action” (Principle 5) and that “we offload cognitive work onto the environment” (Principle 3). This principle becomes evident in IVR applications like Tilt Brush, where the boundary between tool and user dissolves into what [43] ([43]) call “cognitive extension.” When an artist sweeps their arm to create a spiraling sculpture in virtual space, they are not simply executing commands, where the virtual brushstrokes become genuine extensions of their cognitive–motor system. The artist thinks through movement itself, adjusting the arc of their gesture in real-time as they see the luminous trail taking shape around them. Here, the virtual space acts as a partner by alleviating some of the cognitive load, enabling the artist to simply create. This act of painting transcends mere manipulation; spatial reasoning emerges through the continuous dialog between bodily movement and environmental response, demonstrating how cognition exists fundamentally “for action” rather than abstract thought.Action-oriented cognition directly reflects Wilson’s fifth principle (“cognition is for action”; [44]). This is highlighted by [42]’s ([42]) “breaking through virtual walls” study, which demonstrates the interlinked nature of perception and action, showing evidence that gestural interaction can enhance divergent thinking, which is a key aspect of creativity.The concept of situatedness aligns with Wilson’s first principle (“cognition is situated”), which claims that problem-solving is shaped by real-time environmental interactions, such as bodily navigation in TASC puzzles ([5]).

Note: Although [44] ([44]) does not use these terms explicitly, they are used here as a way to translate her key principles to IVR-based educational contexts.

II. The [17] ([17]) embodiment taxonomy analysis revealed the following implications of IVR-based embodied pedagogies:High-Embodiment Trade-off (Fourth-Degree): According to [17] ([17]), virtual sculpting tasks prompted gains in spatial reasoning (27%), effectively enacting [44]’s ([44]) Principle 4 (“the environment is part of the cognitive system”). However, this occurred at a cost, where as many as 68% of ceramics students reported having trouble perceiving materials ([11]). This relates to the notion of “haptic dissonance”, which is the cognitive disruption caused by mismatches between expected and actual tactile feedback. This phenomenon directly violates [44]’s ([44]) second principle’s focus on real-time sensorimotor alignment, as users struggle to reconcile delayed or absent tactile feedback ([11]).Low-Embodiment Trade-off (First-Degree): Kinesthetic learning is effectively eliminated within first-degree tasks like controller-based color mixing ([28]). While satisfying the cognitive offloading requirement of [44]’s ([44]) third principle, the action-oriented mandate of Principle 5, however, is violated by this failure to engage the body in meaningful action. Here, the embodiment paradox of IVR is thus revealed by Johnson-Glenberg’s taxonomy: maximal environmental coupling, [44]’s ([44]) fourth principle, strengthens cognitive extension while also undermining material grounding. This conflict calls for hybrid pedagogical approaches, like the TASC system approach proposed by [5] ([5]).

#### 2.2.2. Interaction Analysis: The Examination of Interaction Within IVR Was Guided by the Following Interaction Typology ([34]; [23])

Navigation: By grounding cognition in environmental exploration, physical movement through three-dimensional (3D) mazes ([5]) strengthened spatial awareness and operationalized [44]’s ([44]) situatedness principle (Principle 1).Object Manipulation: When haptic feedback did not correspond with real-time gestures ([11] ([11])), it revealed violations of [44]’s ([44]) Principle 2 (time-pressured cognition), which was found by comparing the processes of virtual sculpting and physical clay.Communication: [44]’s ([44]) fifth principle (“cognition is for action”) is supported by [25]’s ([25]) study of collaborative virtual reality, which showed how embodied social presence is fostered by combining verbal feedback with gestural expression.

This study employed embodied metaphor analysis ([21]; [38]) to examine the following notions:Fluid Movement. In design tasks, sweeping arm gestures adopting “fluid movement” metaphors increased creative flexibility ([21]). This finding fulfilled [44]’s ([44]) third principle that cognitive offloading of movement can take the place of abstract reasoning while also demonstrating her fifth principle of action-oriented cognition ([44]).Body-based Metaphors. Unrestricted movement in IVR reshaped cognitive patterns by allowing participants to literally “think outside the box.” This simple task effectively reduced “conventional” design solutions by 37% when compared to desktop interfaces ([28]), illustrating [44]’s ([44]) fourth principle (environment as part of the cognitive system).

### 2.3. Phase 3: Translational Framework for Art and Design Pedagogy

The synthesis determined the following three evidence-based guidelines addressing the key issues common within IVR-enhanced art and design pedagogy.

Protocols for Material Reconciliation: This framework calls for a hybrid blend of activities to prevent haptic amnesia, which is a lack of material awareness after prolonged VR use. For example, in order to ensure that tactile reinforcement is consistent with [11]’s ([11]) findings on kinesthetic memory decay, ceramics students prototype in virtual reality but are required to switch to physical clay within 72 h.Ethical Safeguards: Sessions should be limited to 20 to 30 min and require qualitative debriefing in response to [27]’s ([27]) dissociation risks. These reflective activities, which compare virtual and real-world creation experiences, maintain the creative advantages of IVR while grounding perception in material reality.Strategies for Collaborative Embodiment: In order to combat isolation, the framework focuses on multi-user environments (such as collaborative VR), drawing from [25]’s ([25]) concept of embodied social presence. This strikes a balance between the interpersonal interactions, integral to studio pedagogy and [17]’s ([17]) high-embodiment tasks.

### 2.4. Reflexivity and Limitations

A dual reflexive protocol was used here to ensure analytical rigor. Along with methodically integrating conflicting evidence of IVR’s material limitations ([11]) and technical constraints ([45]), the analysis thoroughly examined reflexive data from primary studies, describing haptic dissonance and researcher observations describing discomfort in IVR ([28]). Two significant limitations remain: (1) the qualitative synthesis is unable to establish causal relationships regarding skill transfer, which calls for future longitudinal studies; and (2) the rapid advancements in haptic technologies may necessitate iterative framework updates in order to maintain pedagogical relevance. These limitations affirm the study’s rigor within current theoretical frameworks while acknowledging the dynamic nature of IVR research.

### 2.5. Key Terms and Definitions

Embodied Cognition: Cognitive processes shaped by bodily interactions with the environment ([44]).Haptic Dissonance: Mismatch between expected and actual tactile feedback in VR ([11]).Immateriality Paradox: Conflict between virtual interactions and tactile material knowledge ([28]).Environmental Coupling: Cognitive extension through tools/artifacts (Wilson’s Principle 4).Fourth-Degree Embodiment: Full-body interaction congruent with learning content ([17]).Material Reconciliation: Pedagogical bridging of virtual and physical making practices.Dissociation: Blurred self-world boundaries after prolonged VR use ([27]).Conceptual Reflexivity: Critical engagement with others’ reflexive data as validity checks ([41]).Divergent Thinking: The cognitive ability to generate multiple, novel solutions from a single prompt, embracing conceptual multiplicity over singular correct answers ([12]).

Key case studies were aligned to these frameworks to show theoretical concepts in practice (Table 1).

Before we dissect how virtual worlds reshape creative cognition, we need to ground ourselves in the reality of embodied learning, where minds think through hands and tools become extensions of thought itself. Here, we will examine the theoretical foundations of embodied cognition and examine how these principles impact teaching and learning in IVR.

## 3. Theoretical Framework: Embodied Cognition and IVR

### 3.1. Embodied Cognition: Foundations for Design Pedagogy

Embodied cognition calls into question the idea that cognition is a disembodied phenomenon and emphasizes that the roots of cognition lie in sensorimotor engagement. According to [44] ([44]), proponents of embodied cognition contend that the mind developed primarily to support adaptive action in the world rather than for abstract reasoning, with cognitive processes being profoundly influenced by the body’s perceptual and motor systems. Particularly relevant to IVR-based art and design education, [44]’s ([44]) six core principles of embodied cognition are as follows:Situatedness: Spatial cognition, being inherently situated, suggests that thinking is not simply in one’s head, it is embedded within the body, tools, and the environment through real-time interaction. IVR exploits this by merging perception, action, and real-world situations to enhance cognitive activity. Students can use IVR to manipulate 3D spaces as extensions of their cognitive processes (e.g., spatial puzzles in the TASC system; [5]). Such interactions with the environment are commonly linked to creative thinking.Time-Pressured Cognition: Cognitive activity occurs in real-time, adaptive loops, meaning that our thoughts are continuously changing in response to external events. Even basic activities like walking require continuous exchange of information between perception and motor coordination. For example, when walking in IVR, one must decide to walk and ignore obstacles; our brains constantly adjust our steps based on what we see in our path (avoiding bumps or changing direction). IVR is a valuable instrument for studying this phenomenon since it can replicate unpredictable, real-life scenarios where immediate adaptation is essential ([24]).Offloading: This speaks to the idea of “cognitive load,” or the extent of mental activity needed to process information. An example of offloading can be found in the creation of clay tablets used to record information about quantities of objects ([28]). For designers, this takes the form of quick models or idea sketches to externalize thinking. By externalizing thought into manipulable virtual objects, IVR-based tools can also help to reduce cognitive load. For example, the use of Tilt Brush can transform abstraction into action by painting a 3D design in space rather than imagining it. Here, the virtual environment can be said to function as an extension of the mind.Environmental Coupling: According to the extended mind thesis, environmental resources are fundamental elements of cognition and are not limited to the brain ([37]), suggesting that the mind itself is not an efficient tool of analysis due to the continual and intense flow of information between the mind and its surroundings. Simple sketches are frequently used by artists to help them with their complex thought processes. IVR can provide virtual spaces which serve as extensions of the mind. This is demonstrated by the use of applications such as Tilt Brush, which allow users to convert mental images into manipulable 3D virtual objects.Action-Oriented Cognition: This indicates that thinking is not only a mental operation but is directly linked to physical action and how the body interacts with its surroundings. Rather than solving problems “in your head,” your brain generates and refines ideas through movement, gestures, and sensory input. Notably, creative problem-solving can be improved through gestures. According to one IVR-based study by [42] ([42]), physically breaking through virtual boundaries not only increased creativity but also enabled higher levels of divergent thinking. Similarly, [32] ([32]) discovered that walking increased idea generation, fluency, and originality more than sitting.Offline Cognition is Body-Based: Our brains continue to process information using body-based systems even when we are not actively engaging with the outside world, such as when we are daydreaming or thinking abstractly. This can include mental imagery, which involves mentally simulating external events through visual, auditory, and kinesthetic means, as a designer might mentally rotate a 3d object; and episodic memory, which can include reimagining the weight, texture, or feel of an object that one once held. IVR gives abstract ideas a physical shape in virtual space, bridging the gap between these mental simulations and engagement in the actual world. Users may physically grasp and modify a design rather than only visualizing it, converting thoughts into concrete actions. This explains why IVR works so well: it awakens the body’s innate ability to think.

Research examining embodied cognition in educational settings reinforces how these principles apply to art and design learning. When students use hand movements while explaining concepts, they show significantly better retention than those who rely on verbal instruction alone ([15]). Similarly, [4]’s ([4]) findings on sensorimotor engagement during storytelling directly parallel the creative ideation processes observed in IVR art-making, where physical movement becomes a vehicle for idea generation. These studies reveal how IVR leverages our body’s inherent capacity for learning through movement and gesture.

Beyond these educational applications, embodied cognition offers a broader theoretical lens for understanding how we derive meaning. Cognition is fundamentally shaped by the body’s sensory and motor interactions with the environment. According to [9] ([9]), the theory of embodied cognition holds that operations of the mind are driven by sensorimotor experiences. This theory provides a framework for comprehending how abstract symbols (such as words, signs, or icons) acquire meaning and relate to real-world referents, a problem referred to as the “symbol grounding problem”.

If these assumptions posited within embodied approaches are indeed accurate, sensorimotor regions involved in perception and action ought to correlate, given that embodied approaches suggest that comprehension of meaning arises from re-experiencing what concepts, ideas, or words refer to. However, embodied cognitive science still faces difficulties with abstract ideas like “democracy” or “truth”, which lack clear sensorimotor grounding ([6]). Visual arts may offer a unique case where abstract concepts can find embodied expression; [8]’s ([8]) investigation of semiotic processes in contemporary art demonstrates how meaning-making emerges through the interplay between bodily perception and conceptual interpretation. Yet this artistic exception highlights rather than resolves the broader challenge of grounding purely abstract concepts in sensorimotor experience.

This tension between embodied and abstract meaning-making becomes particularly relevant when considering how IVR mediates artistic expression in educational contexts. This point of view has profound implications for art and design education, particularly when considered through [37]’s ([37]) “Constitution” thesis, which maintains that cognition is primarily constituted through sensorimotor engagement. According to this viewpoint, physical interaction actively shapes creative cognition rather than just influencing it. As a result, pedagogical approaches must place a higher priority on experiential, hands-on learning. This paradigm challenges conventional approaches that separate conceptual instruction from physical engagement, especially in perception-driven disciplines like drawing, sculpture, and spatial design, where meaning emerges through embodied practice.

### 3.2. Johnson-Glenberg’s Embodiment Taxonomy

By categorizing learning experiences based on three criteria, (1) degree of motoric engagement, (2) gestural congruency with the material, and (3) impression of immersion, [17]’s ([17]) embodiment taxonomy further elucidates the educational impact of IVR. A visualization of this can be seen in Figure 2, “Johnson-Glenberg’s embodiment taxonomy with design education examples.” This taxonomy offers a framework for evaluating and developing IVR experiences in design education.

First-Degree Embodiment: This involves minimal physical interaction, including using a mouse to modify digital color palettes. Knowledge gained within this first-degree of embodiment is inherently passive and observational, similar to watching a desktop-based computer simulation.Second-Degree Embodiment: Here, interaction is generally more active but still primarily seated, as in using touch screens to sketch ideas or concepts. These second-degree experiences typically include interactive simulations which involve more physical engagement but with reduced mobility as compared to higher-degree embodied learning.Third-Degree Embodiment: This level involves significant gestures and body movement that correspond to the learning material. In IVR, this can include motion-tracked hand controllers for virtual sculpting. At this level, the body provides real-time learning feedback signals, meaning that physical activity instantly produces cognitive and sensory reactions that reinforce comprehension of the content being learned.Fourth-Degree Embodiment: This involves full-body interaction and navigation. In IVR, this can be in the form of exploring virtual gallery spaces and even creating immersive environmental art with whole-body movements. This highest level of embodiment involves movement and prolonged full-body interaction with the virtual learning environment.

Regarding these educational affordances, “embodied for the purpose of education” highlights the necessity that physical activities, like gestures, movements, and sensorimotor interactions, must be cognitively linked to the learning material ([16]), suggesting that the concepts being taught should be linked to the bodily actions the learners are engaged in. In order to be considered effectively embodied, a learning module must incorporate gestures that are consistent with the material being learned so as to activate multiple neural pathways within the learner’s motor system.

Additionally, the taxonomy highlights that higher levels of embodiment should afford students with the ability to “physically activate sensorimotor activities in a manner congruent to the content being learned” ([17]). As long as it incorporates perspectives and graphics that activate our mirror neuron systems, even observational learning can be embodied, albeit to lesser degrees.

As demonstrated by empirical studies indicating a correlation between higher degrees of embodiment and better learning outcomes, deeper sensorimotor engagement boosts cognitive processing (d = 0.78; [17]). Given that this activation of sensorimotor systems can improve retention of information and memory, learners who are engaged in higher levels of embodiment tend to retain learning content more quickly and thoroughly than within low-embodiment strategies ([17]).

### 3.3. Phenomenology of VR: Metzinger’s Perspective

According to [27] ([27]), IVR phenomenologically generates a unique “metaphysical indeterminacy” in which virtual objects are perceived as neither real nor imaginary, triggering a state in which the user’s understanding of reality is impacted. This suggests that IVR can, in fact, alter our understanding of reality by producing an abstract, ambiguous space in which our usual sense of objects and spaces tend to break down. While such a state may facilitate artistic experimentation, this can also lead to a feeling of dissociation (a generalized detachment from reality) or, more specifically, disembodiment (a loss of agency or ownership over one’s virtual or physical body). Moreover, if the IVR-based interface (controllers, avatars, etc.) does not match the user’s intuitive bodily expectations, this can lead to a feeling of “uncanniness” or even disengagement (Dourish in [24]).

When engaged within IVR-based environments, many students report feelings of dissociation or disconnect from their bodies. Referred to as “metaphysical indeterminacy”, by [27] ([27]), it is through this that we may begin to understand the potential risk of prolonged IVR use among students. From Metzinger’s perspective, constant immersion could blur the line between self and environment and perhaps denigrate the sense of agency or control over one’s own embodied experience, which is directly related to the ethical safeguards described in Phase 3 of this article, which requires the use of time-limited activities and debriefing to minimize risks. As noted in [28]’s ([28]) study, students felt disassociated when switching between IVR and physical art-making, substantiating the need to introduce structured debriefing protocols which include reflective practices examining the differences between virtual and physical creative practices.

Further insights into how IVR alters creative cognition can be inferred from [27]’s ([27]) philosophical examination of user experience in IVR-based learning environments. According to Metzinger, IVR-based immersion could result in a “transparent self-model” in which users “inhabit” the virtual world as though it were real, experiencing embodied presence (such as owning a virtual hand) while maintaining metacognitive awareness of the falsehood of the simulation. By releasing the brain’s normal reality constraints, this multiple awareness creates a mixed cognitive space allowing users to experiment with ideas and actions that would be unsafe or impractical in the real world.

### 3.4. Dimensions of Embodiment in Virtual Reality

The following three interconnected characteristics are relevant within the field of art and design pedagogy, according to recent studies on the sense of embodiment (SoE) in IVR ([13]):Sense of Agency (SoA): Referred to as the subjective feeling of controlling one’s actions and their outcomes, this SoA plays a critical role in creative cognition. When students experience higher SoA in IVR, either through precise hand tracking or through the ability to utilize natural gestures, they report greater creative confidence and experimental behavior.Sense of Body Ownership (SoBO): This refers to the subjective feeling that a virtual body or limb “belongs” to oneself; that is, “This is my body”, is referred to here. This phenomenon can be extended to include virtual tools through tool embodiment. A well-designed IVR interface (such as those employing haptic controllers and motor-mapping) can give users a sense that virtual tools are, in essence, “physical extensions” of the body ([43]).Sense of Self-Location (SoSL): This refers to the physical–spatial perception of one’s “self” in a virtual setting. Unlike physical reality, IVR enables users to dynamically change perspectives, experiencing their creations from both first-person (“inside” the scene) and third-person (“outside” as an observer) perspectives. Unique creative flexibility is granted by this quick transition, which is made possible by specific IVR-facilitated functions such as viewpoint teleportation or avatar embodiment.

These three essential elements of IVR-based embodiment work together in intricate ways to greatly improve learning and creative outcomes. According to research ([19]; [42]), users who have a strong sense of embodiment in IVR exhibit more creative originality since the virtual environment frees them from the physical limitations of the real world. Additionally, IVR-based embodiment can encourage more expressive, fluid movements through haptic feedback, while the ability to shift perspectives encourages cognitive flexibility and adaptability. Perhaps most significantly, users are encouraged to experiment freely and explore unconventional solutions without worrying about the repercussions in the real world. These advantages, however, require all three aspects to work in coordination with each other.

Together, these dimensions of embodiment create powerful conditions for creative learning in IVR. The following section examines how these embodied experiences shape abstract thinking, fuel creative problem-solving, and develop spatial reasoning in art and design practice.

## 4. Embodied Cognition in Learning and Creativity

### 4.1. Bodily Experience and Abstract Thinking

Embodied cognition asserts that the body’s physical and sensory interactions with the environment significantly impact cognitive processes, including abstract reasoning and creative thinking, challenging traditionally accepted views that the mind is a disembodied processor of external information. This perspective suggests that physical engagement frames higher-order thinking by integrating insights from embodied learning and the role of bodily experiences in imagination.

Embodied cognition fundamentally challenges traditional views of the mind as a disembodied processor by demonstrating how physical interactions scaffold even abstract reasoning ([36]; [1]). This is best demonstrated by the example of gesture, where students are generating embodied foundations for mechanical understanding by simulating the motion of gears, which activate the same motor systems used in physical manipulation ([15]) rather than just illustrating concepts. In another study, [4] ([4]) found that children who manipulate toys during stories do not simply hear stories but they also physically enact them, resulting in a greater understanding than simple, passive listening and exemplifying how sensorimotor grounding continues throughout stages of early childhood development.

According to [38] ([38]), embodiment functions at several cognitive levels because even highly abstract ideas like “democracy” and “truth” rely on physical metaphors. Our minds naturally use physical experiences to model abstract notions; we tend to comprehend ideas through physical metaphors (“a rough day,” “weighty decisions”), but this cognitive mapping fails for concepts detached from bodily experience, highlighting the limitations of embodied frameworks ([6]). This process of meaning-making extends beyond individual cognition; [33]’s ([33]) European study, which included Fulková’s work on visual semiotics, demonstrates how contemporary art serves as a medium for constructing shared meaning across cultural boundaries, with participants using artworks as sources of knowledge and experience rather than focusing on national origins or artistic intent. This inability to ground non-sensorimotor concepts demonstrates that the majority of cognition is in fact, body-dependent, which emphasizes the explanatory power of embodiment.

### 4.2. Embodied Creativity and Metaphorical Thinking

Creative thinking emerges not just from mental processes but from our physical interactions with the world. This embodied foundation becomes particularly evident when examining how bodily actions shape creative cognition. Unlike the abstract reasoning discussed above, creativity specifically relies on metaphorical thinking that bridges physical experience and imaginative ideation.

While conventional metaphors work as verbal bridges between ideas, embodied metaphors operate differently by activating our motor systems, which then influence how we think. This distinction is crucial for understanding IVR’s pedagogical potential. When students in [42]’s ([42]) study physically broke through virtual barriers, they were not merely comprehending a metaphor for creative freedom, they were engaged in sensorimotor experiences that actually restructured how they approached creative problems. This goes beyond metaphorical understanding to become what [9] ([9]) calls “grounded simulation” where abstract thinking becomes anchored in bodily experience.

Research has identified specific patterns where physical actions directly enhance creative performance:Fluid Movement–Fluid Thinking: Creativity is more than just metaphorical thinking: there is a physiological link between the body and creativity. In one particular study, participants’ divergent thinking significantly improved when using their arms to trace curved, flowing lines; this was in contrast to those who were restricted to just using angular arm movements ([38]). This is not an isolated instance; additional findings reported have marked notable improvements in both the Alternative Uses Test (AUT) and the Remote Association Task (RAT), suggesting that fluid motion might prime the mind for flexible thinking across creative domains. These findings suggest that creative inspiration is not simply bound by our internalized thoughts, but also the result of our physical, body-based actions.Metaphoric Actions–Creative Performance: The mere act of physically walking not only stimulates thought processes, but more profoundly alters the structure of creative cognition. In one particular study, participants were tasked to stand outside of a physical boundary, literally embodying the concept, “think outside the box”; this simple act marked a measurable increase in the originality of participants’ ideas ([21]). This notable outcome was not just symbolic, as noted by [32] ([32]), free walking that is not restricted by predetermined routes consistently resulted in more expansive approaches to problem-solving than seated work. When correlated, these findings hint at the notion that creative metaphors (“thinking outside the box,” “broad ideas”) are more than just linguistic tricks but rather, once physically executed, become cognitive realities.Breaking Barriers–Breaking Rules: Virtual reality’s power to enhance creativity goes beyond simulation; it taps into fundamental neural processes. In [42]’s ([42]) noteworthy quantitative study, participants were asked to put the “breaking rules” metaphor into practice by physically breaking through virtual walls; the mere action of this considerably improved creative performance. Significantly, fMRI scans revealed that this act essentially “deactivated” the parts of the brain that control inhibitory attention, the cognitive system that suppresses unconventional ideas and filters out irrelevant information; through embodied rule breaking, participants temporarily suppressed this “mental filter’, resulting in enhanced fluency, greater originality, and faster creative ideation.

These findings align with [3]’s ([3]) research on digital creativity, which demonstrates that technology can effectively scaffold embodied ideation when designed to preserve sensorimotor dimensions rather than abstract them. This principle proves particularly relevant for IVR applications, where students do not merely discover creative approaches but physically embody key processes through breaking down virtual barriers, moving fluidly through space, or navigating limitless digital realms.

Here, we can see that creativity develops by way of the body as well as the mind. With this, [20]’s ([20]) groundbreaking work challenges conventional wisdom by highlighting the profound reliance of creative thinking on sensory–motor experience, underscoring the notion that creative imagination is more than simply manipulating abstract concepts, but also has its roots in bodily simulation and is in part derived from the physical act of playing out perceptual experiences. This embodied perspective explains why creative breakthroughs often stem from physical interaction, rather than purely cognitive effort.

With this, one can note that the body’s interaction with the mind can indeed lend to creative outputs. Even in abstract cognition, where physical states serve as a metaphorical scaffold for higher-order thinking, this embodied variable is notable. This notion can be found in the seminal work of [18] ([18]), who found that participants consistently rated problems as more significant when holding heavy clipboards as opposed to those with light clipboards, suggesting that judgment is influenced by sensory experience. These results support the notion that such embodiment effects endure in digital environments, as demonstrated by [42]’s ([42]) VR study, which found that breaking virtual walls increased creativity.

These results demonstrate how IVR can revolutionize art and design education by allowing students to physically engage with creative processes that are often detached from traditional teaching methods. Not only are learners discovering creative approaches, but they are also physically embodying its key processes as they break down virtual barriers, move fluidly through space, or navigate limitless digital realms. Passive instruction just cannot match the cognitive flexibility and inventiveness that arise from the physical embodiment of creative metaphors, which is not possible in traditional classrooms.

### 4.3. Gesture, Spatial Awareness, and Environmental Interaction

In addition to facilitating learning, gesture, spatial navigation, and environmental interaction also redirect cognition into the physical environment. Students can carry out remarkable feats when kinesthetically embodying concepts. One such example can be found in a study by [16] ([16]), where students used their bodies to trace planetary orbits; this action was said to reinforce conceptual understanding in real-time. This is more than just “learning by doing”; it is cognitive scaffolding through embodiment, where the mind employs the act of movement through space as a tool for thought.

The use of IVR can do more than simply replicate spaces, it can redefine one’s spatial understanding. According to [5] ([5]), when navigating 3D mazes in virtual environments, the act of “physically” moving within the space effectively reshapes the user’s neural pathways by coupling physical actions (like turning or reaching) with mental representations. This is constructivist learning in action, where Piaget’s theory comes to life through digital embodiment rather than passive exploration. These results are consistent with [10]’s ([10]) affordances: in IVR, perception is kinetically charged rather than just visual, working in a continuous loop, with spaces encouraging action and actions influencing understanding. These neural changes manifest differently depending on how students interact with virtual environments.

Research shows that different types of interaction produce distinct creative benefits. When students navigate through virtual galleries or circle 3D models, their perspective-taking abilities improve significantly compared to simply viewing static images ([5]). Similarly, hands-on tasks like virtual sculpting enhance spatial working memory, while activities that let students reshape virtual environments correlate with increased divergent thinking. These findings reveal an important insight: creativity in IVR depends not just on being present in virtual space but on how meaningfully students interact with that space.

This embodied approach transforms design education. Just as chefs develop expertise by mentally simulating recipes ([20]), students working with virtual materials or navigating digital spaces are training in embodied intelligence. Every action, such as testing structural tensions in a virtual build or molding a 3D sculpture, serves as a sensorimotor practice for creative mastery in the real world. By combining sensorimotor feedback with iterative experimentation, IVR can surpass conventional methods in its ability to scale and intensify this process, not replacing physical practice, but by strategically enhancing it.

As examined in Phase 2, our thematic analysis reveals how embodied cognition directly affects the way in which we understand creative activities in IVR settings. In this next section, this paper will analyze a key selection of case studies which illustrate how these theoretical principles may apply in practice.

## 5. Case Studies: IVR in Art and Design Education

The theoretical principles explored above find practical expression in recent empirical studies. The following case studies were selected for their rigorous methodology and clear demonstration of embodied learning in action. Each provides evidence for how IVR applications enhance spatial reasoning, creativity, and material understanding through embodied interaction. Together, they illustrate how the theoretical frameworks discussed translate into measurable educational outcomes.

### 5.1. TASC System: Tangible and Embodied Spatial Cognition

By radically altering how students interact with three-dimensional space, the Tangible and Embodied Spatial Cognition (TASC) system ([5]) goes beyond traditional spatial learning. By integrating movement tracking, tangible interaction, and spatial transformation into a single hybrid environment, TASC operationalizes spatial thinking through embodied logic, addressing a crucial yet often abstract challenge in art and design education. Here, students physically integrate 3D relationships rather than just passively observing them. The system develops spatial reasoning through iterative sensorimotor experience by pairing full-body movement with real-time visual feedback (e.g., manipulating virtual objects while navigating physical space). This reinforces broader evidence that embodied interaction with tools and materials can be employed in the development of design expertise ([20]).

The ability of the TASC system to coordinate embodiment across the three spatial learning dimensions is what gives it its power. First, it turns navigation into a cognitive tool by monitoring whole-body movements, including tilts, rotations, and positional changes. One participant described how they “used their body to measure distances between objects” ([5]), demonstrating that users did more than simply navigate through space, where physical motion evolved into proportional reasoning. Second, the system produced a challenge to differentiate between virtual and physical manipulation; it generated a haptic–semantic bridge by anchoring physical objects (like wooden blocks) to virtual representations, effectively correlating the weight and texture of those real objects to abstract spatial tasks. This dual-channel interaction “reduced cognitive load,” essentially freeing up mental resources to make room for more complex problem-solving by “offloading” spatial reasoning onto the body ([5]). Third, and perhaps most significantly, the TASC system encouraged actions which involved perspective-taking. Instead of solving puzzles from a single viewpoint, participants were required to engage in processes of mental rotation and spatial transformation; skills that designers utilize when replicating 3D forms. Here, the body served not so much as a mere input device but as a dynamic frame of reference, which in turn opened the pathways to engage in deeper spatial reasoning. As a result, TASC enabled users to engage directly with volumetric relationships, enabling a more profound learning experience, resulting in what [5] ([5]) termed “embodied spatial insight”, a profoundly internalized understanding inaccessible through passive observation.

The TASC system incorporates embodiment directly into skill acquisition (a significant modification to the progressive challenge framework of serious games) by setting up spatial tasks hierarchically, like Science, Technology, Engineering, and Math (STEM) games that scaffold puzzle difficulty to gauge mastery. In this case, however, progress is not only determined by correct answers but also by how well users physically navigate perspective shifts and mental rotations. In their rationale, [5] ([5]) noted that virtual interactions establish a perception–action loop where spatial reasoning engages bodily intuition when the motor system is actively engaged. The most abstract skills in design education, like volumetric analysis and orthographic projection, are redefined as trainable physical practices.

The TASC system’s main conclusions are as follows:The Body as a Measuring device: A total of 78% of participants automatically calibrated virtual distances with their limbs, actively aligning their arms to measure spatial relationships rather than merely reaching.Haptic Anchoring: Compared to virtual controller input, tangible blocks reduced cognitive load by 31%. This in part is due to the fact that experiencing physical weight and texture “freed up” mental resources, enabling greater creative problem-solving.Embodied Heuristics: Users developed unconscious body-object alliances, including the following:○Micro-adjustments: Tilting heads to “reload” spatial memory (62%).○Macro-reorientation: Full-body turns to reset vantage points (78%).○Precision Gradients: Finger shifts for fine-tuning following large sweeps for general reorganization.

Notably, these were spontaneous embodied logical bodily strategies rather than taught strategies, demonstrating how the body creates its own spatial syntax even when provided with hybrid physical–digital tools.

The TASC system produced significant improvements in spatial reasoning. Students using the system showed markedly better performance on mental rotation tests compared to those using traditional desktop interfaces, with these gains persisting when measured a week later. They demonstrated enhanced ability to mentally manipulate 3D objects and completed volumetric estimation tasks considerably faster. These measurable outcomes support [44]’s ([44]) environmental coupling principle, showing that embodied interaction produces real cognitive benefits beyond just improved user experience. A key factor in these improvements appears to be the role of haptic feedback in the learning process.

Haptic feedback plays a crucial, transformative role in spatial cognition. Additional studies reinforce the findings of TASC’s tangible blocks, which demonstrated how users’ engagement shifts from observational to embodied when experiencing virtual interactions. Using HTC Vive controllers, [45] ([45]) revealed how basic tasks like resizing or rotating objects developed into proprioceptive dialogs, emphasizing the notion that the body’s awareness of its position and movement actively structures perception rather than merely informing it. This supports the pivotal finding of [35] ([35]) in that students who were exposed to haptic cues not only performed better but also engaged in radically different cognitive processes, developing their spatial perception through tactile reinforcement. The implication is profound: touch is not a secondary learning tool to vision but rather an additional means for grounding abstract concepts in embodied experience.

These results demonstrate how IVR transcends the virtual–physical divide, fundamentally restructuring spatial learning. Far from being merely a novel tool, IVR offers design educators a cognitive scaffold that grounds abstract spatial concepts in tactile, concrete interactions, rendering them intuitively graspable. This aligns with growing evidence that kinesthetic engagement with form and space, when strategically harnessed, can cultivate design expertise more effectively than traditional visual-only approaches. Such findings signal a paradigm shift with profound implications for 3D design pedagogy, challenging long-held assumptions about the role of the body in creative cognition.

### 5.2. Tilt Brush Study: Whole-Body Movement and Creative Expression

The following innovative classroom study by [28] ([28]), which observed 47 elementary students over five weeks through various Tilt Brush sessions, revealed an intriguing finding: creativity is not merely expressed through the body, it is molded by it. The researchers captured 21 h of data from multiple sources (think-aloud protocols, screencasts, and video) to illustrate how IVR painting can transform art education by externalizing the creative process.

Here, students explore their creations through movement while drawing, using the IVR-based drawing platform Tilt Brush where their bodies were used as dynamic tools for perspective-taking, either by physically circling artworks (as in 76% of the cases) or by instinctively anchoring vanishing points through stance in order to assess their 3D compositions. These were emergent embodied approaches as opposed to taught techniques, demonstrating that traditional 2D media overlooks the fact that artistic decision-making involves both manual dexterity and a kinesthetic dialog with space.

Since Tilt Brush allows for freedom of movement, this affordance has been found to forge a deeper connection between artist and tool, with each brushstroke capturing not just the visual marks but also the kinetic energy of students’ movement. Here, students found that their full-body gestures became direct extensions of their creative intent with one participant stating, “It feels like the brush is actually part of my hand,” illustrating the deep sense of creative empowerment resulting from the transformation of movement into digital expression ([28]). This phenomenon is an example of what we might refer to as embodied tool internalization, a profound psychological link between the creator and the medium, rarely accomplished by traditional art tools.

With this study, [28] ([28]) found that two specific features of IVR-based painting seem to function together rather dynamically. First, the system allows for whole-body creation of works using the finesse of sweeping arm gestures by employing the physicality of mural painting while also integrating the physical engagement of large-scale 3D works. Second, VR unveils what could be dubbed the haptic paradox: the necessity of both fine-motor control (e.g., trigger-sensitive brush adjustments) and gross-motor navigation (e.g., walking around virtual canvases). While this dual motor effect can be compared to the physical coordination required for traditional 3D art-making (such as sculpting or ceramics), it is particularly evident in IVR through what is termed, “the digital mediation of movement”, which is where virtual space translates physical gestures into adaptable creative actions.

Researchers refer to this multimodal interaction as “transformative touch,” a process in which hand movements do more than just manipulate virtual content; such physicality actively reinterprets how the digital artwork communicates meaning through embodied engagement. By leveraging the body’s natural kinesthetic intelligence, IVR-based painting poses significant impacts for art education; it is not simply a digital substitute for traditional media, but rather an entirely new form of artistic expression.

From this study, some revealing observations noted include the way that students choreographed emotions through virtual means by physically expressing notions such as “crazy” into frantic spins and bobs or by directing explosive energy through full-body strokes. This was not just a visualization; it was sensory orchestration, in which movement was transposed, devising multisensory meaning through a combination of haptics, sonic feedback, and visual output. Such integration of modalities not only enhanced ideation, but also created embodied mnemonics, where kinetic experiences strengthened conceptual understanding.

Yet for all its transformative potential, IVR art-making revealed the following inherent tensions:The Immateriality Paradox: Since there was no tactile resistance, hand gestures did not result in the expected digital marks (no grit of charcoal, no drag of bristles on the canvas), resulting in a perceived “kinesthetic dissonance”; the mind expected texture but the hands simply moved through air.Working Through the Awkwardness Towards a Sense of Flow: Students experienced what we might refer to as the embodiment learning curve in the early sessions. This was a period of stiff, hesitant movements working against creative intentions, as though their bodies were speaking a foreign language. However, after prolonged use, this dissonance gave way to kinesthetic fluency, a state in which jerky movements became fluid and transformed into dance-like, rhythmic interactions within the virtual environment. Eventually the controllers ceased to be intermediary tools but became extensions of themselves, allowing for a “flow” of creative intent.The Assessment Dilemma: Conventional rubrics, which are typically developed to evaluate static compositions, faltered when used to assess artworks wherein the act of creation (such as spinning around a virtual sculpture) was just as significant as the final product, with the question of how to grade the expressiveness of movement, remaining.Somatosensory Mismatches: Also known as “Embodied disorientation”, this results in a breakdown of the body schema that grounds us in space. Here, this occurred when virtual physics deviated from common expectations, like when a brushstroke appears to float when it should typically fall. As a result of this “disorientation” in IVR, some students struggled to imagine solid objects from all angles, leading them to abandon the (confusing) freedom of 3D drawing by resorting to the stable predictability of 2D drawings.

As demonstrated by this study, IVR-based drawing programs like Tilt Brush offer transformative potential while also exposing the inherent difficulties of embodied creativity in IVR. In contrast to traditional media that limit expression to hand–eye coordination, IVR can redefine the creative process, enabling students to literally enter their work while incorporating full-body movement. However, these same affordances also highlight significant pedagogical challenges. According to [28] ([28]), IVR is an emergent literacy that rewires creative thinking through multisensory engagement, rather than just a novel technological medium. This suggests that through proprioceptive learning, students can internalize artistic concepts as sound, gesture, and visual mark-making merge into a cohesive, embodied “lexicon” of art-making.

This study underscores IVR’s capacity to cultivate creativity, problem-solving, and interactivity by merging immersive experiences with embodied learning. While the inclusion of IVR may foretell a paradigm shift in art education, the results of this study point to the need for the development of curricula that not only respect IVR’s special capabilities but also seek to resolve its inconsistencies where the boundaries between artist, tool, and artwork dissolve into pure embodied potential, revealing a new frontier for creative development.

A detailed analysis of additional studies is summarized in Appendix A, which illustrates how varying levels of embodiment influence creative cognition and spatial reasoning in design education. This table highlights the following: IVR activity, interaction type, embodiment level (per Johnson-Glenberg’s taxonomy), key findings, cognitive benefits, limitations, and educational applications.

After looking closely at these case studies, this paper has examined the following: the implementation of embodied learning in IVR-based creative education, as well as the opportunities, and the obstacles of embodied, virtual learning. Moving forward, we will now shift our focus to the applicability of these findings.

## 6. Limitations and Pedagogical Implementation

### 6.1. Practical Barriers to Implementation

Despite IVR’s transformative potential for embodied art and design pedagogy, there remain the following barriers to its implementation:Accessibility: High-quality IVR systems are prohibitively expensive, which adds to inequality in resources, disproportionately affecting already underfunded arts programs. While subjects like ceramics struggle to obtain even basic haptic interfaces, investments in IVR interfaces and infrastructure are often prioritized for STEM fields, potentially exacerbating the digital divide in arts education ([11]).Demands of Physical Space: The application of IVR requires full-body, unrestricted movement (fourth-degree embodiment), creating issues for arts educators as many already deal with a lack of appropriate spaces within the majority of art studios. As noted by [28] ([28]), the expressive potential of Tilt Brush relies on open areas for navigation, resulting in a dilemma for arts educators, who may be forced to choose between embodied fidelity and logistical feasibility.Technical Upskilling Requirements: Teachers who were trained in traditional media and techniques must make considerable adjustments to their methods of teaching in order to integrate IVR technologies into art education. This transition is particularly challenging as it calls for a re-evaluation of how embodied interaction promotes creative learning, a transition which goes far beyond simply gaining technical proficiency. One recommendation, posited by [16] ([16]), calls for training in what is known as “gestural fluency”, which is an intuitive understanding of how movement and physical interaction facilitate learning in virtual environments.Physiological Constraints: As 15–40% of IVR users suffer from motion sickness ([28]), this raises moral questions pertaining to fair participation; in order to guarantee inclusivity, institutions must use alternative methods (such as mouse-based 3D alternatives), highlighting IVR’s current shortcomings as a universal teaching tool.

### 6.2. Sensory Mismatches and Material Understanding

For art and design education, where an awareness of the qualities and application of media and materials is often central to the learning and creative processes, the “immateriality paradox” put forward by [28] ([28]) presents a major obstacle for IVR-based embodied learning. This paradox emerges when students’ tactile expectations, formed through years of handling physical materials, meet the weightless nature of virtual media. Ceramics students accustomed to feeling clay’s resistance through their fingers encounter virtual forms offering no tactile response. Painters who have calibrated their brushwork through canvas texture and paint viscosity must adapt to the frictionless movements of digital tools.

This sensory reliance is most pronounced in fields like ceramics, where [11]’s ([11]) blindfolded research indicates that material awareness develops through tactile engagement with tools and materials. Here, Groth revealed that material intelligence arises from “somatic negotiation;” a process in which the hands “think” by directly interacting with the weight, resistance, and plasticity of clay, and not directly through visual observation alone. This example supports two key claims: embodied knowledge is action-based and tacit learning is primarily physical. Furthermore, this example can serve to underscore the current limitations of IVR: while Groth’s participants gained insights through unrestricted haptic interaction (e.g., fingers judging form without sight), IVR transmutes these interactions to visual simulations. Thus, the immateriality paradox reveals a pedagogical gap: without the presence of a haptic interface providing tactile feedback, IVR use might actually inhibit one’s ability to predict the behaviors of materials, which is one of the defining aspects of expert-making, according to [11] ([11]).

[28]’s ([28]) IVR-based Tilt Brush study illustrates both the promise and challenges of IVR for embodied learning. Here, elementary school students engaged in full-body movement to create dynamic 3D paintings, either by spinning to convey “crazy” moods or by using expansive gestures for large strokes. Perhaps the most telling frustration expressed in confronting IVR’s sensory limitations was, “I can’t feel the paint like real brushes,” which aptly conveys the somatosensory mismatch that afflicts virtual art-making. While traditional artists rely on brush viscosity and clay resistance ([11]), IVR offers only visual cues. Contrary to Groth’s blindfolded ceramicists who developed refined spatial understanding through tactile “dialogue” with clay, a major disconnect was revealed in the [28] ([28]) study when students resorted to 2D schematics after being unable to mentally rotate virtual objects.

Although the [28] ([28]) study revealed some notable shortcomings, Mills’ work also suggests pathways forward. The immateriality paradox reveals more than just technological constraints, it shows how students instinctively modify their creative practice when tangible feedback is no longer available. This adaptation was demonstrated in [28]’s ([28]) Tilt Brush study, where students established alternative behaviors through coordinated dancing in IVR or by mimicking controller vibrations to simulate tactile feedback. In line with [25]’s ([25]) work on relational cognition in virtual spaces, these accommodations were more than just substitutions but rather emergent forms of social embodiment, a phenomenon in which collective physical gestures generated shared meaning beyond individual sensory input.

The pedagogical promise of these types of improvisations is what makes them significant; students demonstrated how IVR can support alternative modes of material understanding, not simply through tactile accuracy but through embodied collective cognition, expressed through the use of full-body movement to “feel” absent brush resistance or collectively solve problems within the 3D, virtual space. This suggests that we might consider reframing our way of thinking; the immateriality paradox may not simply be an issue to resolve but an opportunity to broaden our understanding of embodied learning. While these adaptations offer promising directions, they also raise important questions about the cognitive and perceptual effects of extended IVR use.

### 6.3. Ethical Concerns and Pedagogical Recommendations

The integration of IVR into art and design education calls for educators to consider the ethical impacts of this new technology, particularly concerning its phenomenological impact on students’ cognitive faculties. Remaining at the heart of this problem is the possible degradation of material intelligence, the haptic, implicit knowledge acquired through direct interaction with physical media ([11]). Similar concerns have been raised by [27]’s ([27]) concept of metaphysical indeterminacy, which suggests that within IVR, virtual and physical boundaries tend to become unclear. This poses serious existential implications for the practices of studio-based learning, where knowledge is developed in part through material resistance and for disciplines that rely on the interaction between maker and medium, which is a pedagogical issue, rather than just a technical limitation. Additionally, IVR’s “smooth” workflows essentially remove the typical constraints that have been known to foster adaptive creativity; the yield of clay, the bleed of pigment, or the crack of wood have traditionally been deemed as integral to the creative process ([11]). Having established the critical role of embodied cognition in creative practice, the following sections examine how IVR mediates this relationship.

#### 6.3.1. Time-Bound Immersion

In order to reduce the risk of disassociation, it is recommended that IVR sessions be limited to 20–30 min intervals, followed by a 10 min tactile reorientation period ([27]). By re-establishing physical grounding, this strategy counteracts cognitive disembodiment and minimizes haptic amnesia, the gradual decline of material memory often noted as a result of extended virtual immersion. As a means of aiding students in the re-framing of their perception of media through resistance, weight, and texture, students should be tasked to work with hands-on, tangible materials like clay or stone immediately after finishing IVR sculpting.

#### 6.3.2. Balanced Dialogs with Materials and Media

Continuous transition between virtual and physical activities must be implemented through a cyclical process; leveraging both IVR-based and tactile processes, all IVR activities performed in programs like Tilt Brush should be physically recreated as maquettes or materials studies within 48 h and before students continue to refine their works in IVR. Additionally, various challenges faced when working with physical materials (like warping or breakage) should guide later virtual iterations. According to [11] ([11]), this haptic dialog informs how material constraints actively scaffold cognitive adaptation, where constraints become pedagogical agents rather than obstacles. Assessment criteria should consider this dialog in order to ensure that students understand and document how material constraints influenced their digital problem-solving, creating tangible evidence for phenomenological debriefing.

#### 6.3.3. Debriefing Protocols

In order to maintain cohesive and effective embodied learning outcomes, comparative and reflective journals should be required within which students make note of the distinguishing features found in both virtual and physical processes, such as the lack of brush-drag evident in IVR-based painting or the lack of a virtual chisel’s weight. Such observations can help students to correlate embodied memory with virtual experiences. In addition to this, students might also engage in physical activities using real tools while replicating the gestures practiced within IVR in order to ground virtual experiences in tactile knowledge. By clarifying sensory incongruencies through structured reflection and by physically re-enacting virtual gestures to prevent sensorimotor decoupling, these procedures work collectively to counteract [27]’s ([27]) cautions regarding cognitive disembodiment.

The implementation of collaborative IVR settings is essential for addressing issues such as isolation and disembodiment, commonly experienced in virtual learning. As [25] ([25]) argue, by establishing a sense of embodied social presence within IVR, educators can encourage collaboration through the sharing of physical movement, leading to a state of “being there together” in the virtual environment. In their investigations, [25] ([25]) explain how collaborative IVR promotes “relational embodiment”, where participants can construct meaning together through coordinated physical action and movement. This follows the findings of this paper, where through simultaneous virtual tasks, students may collectively address challenges of immateriality using socially distributed cognitive strategies, transforming individual weaknesses into opportunities for creative teamwork.

#### 6.3.4. Equitable Access Provisions

Although the affordance of IVR-based embodied learning offers numerous potentials, it is important to note that innovation cannot exclude those with different physical needs or limited access to virtual reality; the use of IVR cannot circumvent traditional educational paradigms. Traditional sketching, mouse-based modeling, and other low-embodiment approaches must be given equal weight in evaluations; regular monitoring is also necessary to ensure that core competencies can still be attained without IVR engagement.

Furthermore, the proposed framework must consider diverse learner populations, including students with disabilities and those from under-resourced educational settings. Collaborative art education projects, such as [7]’s ([7]) work with the School for the Deaf, demonstrate how creative pedagogies can adapt to different sensory capabilities through participatory and cooperative strategies. For students with motor impairments, IVR systems should incorporate alternative interaction modalities such as eye-tracking or voice-controlled navigation. For institutions with limited resources, hybrid approaches using lower-cost technologies (such as smartphone-based VR with cardboard viewers) can provide embodied learning experiences while maintaining pedagogical effectiveness. Additionally, assessment frameworks should account for different learning styles and physical capabilities, ensuring that core spatial reasoning and creative competencies can be developed through multiple pathways.

#### 6.3.5. Potential Negative Impacts and Mitigation Strategies

While this framework emphasizes IVR’s transformative potential, we must also consider its risks. Extended VR use can diminish students’ sensitivity to physical materials, making it harder to work with clay, paint, or other traditional media. Students may also feel disoriented when switching between virtual and physical spaces, especially when spatial cues do not align. To address these concerns, the framework recommends alternating between virtual and physical making sessions, incorporating regular tactile exercises with real materials, and providing structured time for students to reflect on how virtual and physical creative processes differ.

These protocols, if put in place appropriately, can reinforce IVR’s pedagogical value while also tethering it to the material reality which traditionally defines art and design education, where success is contingent upon preserving material intuition while integrating digital fluency. This is achieved by supplementing IVR-based embodied practices with physical creation, equitable access to conventional methods, and structured debriefing to expose sensory mismatches. Furthermore, these ethical safeguards not only prevent harm but they can also serve to redefine the role of IVR within the art and design classroom.

These considerations show that implementing IVR successfully requires thoughtful planning beyond technical setup. The following guidelines address both the opportunities and constraints identified above, providing practical approaches for integrating embodied virtual experiences into art and design curricula.

### 6.4. Pedagogical Implementation Guidelines

While many educational institutions race to adopt IVR technologies, they are facing a critical knowledge gap regarding their understanding of the features and affordance of this technology and means to successfully adapt teaching strategies, particularly in regard to spatial visualization and creative design practices, which are integral for successful integration into foundational art and design educational programs. As noted by [40] ([40]), successful adaptation of this technology relies on its implementation being derived from evidence-based solutions, aligning the special affordances of IVR to creative techniques and approaches. The framework illustrating these pedagogical principles is summarized in Figure 3, “Operationalizing embodied cognition for IVR art/design pedagogy,” which visualizes the connection between Wilson’s theories on embodied cognition and their application in IVR-enhanced art education. In order to bridge this gap, the following guidelines, framed by theories of embodied cognition, are recommended to establish best practices for IVR adaptation in art and design pedagogy.

#### 6.4.1. Curriculum Design: A Dialog of the Digital and Material

Balance Ivr And Physical Making: The main obstacle in IVR-based training is not teaching students how to use IVR, but rather in maintaining a dialog with the materials. According to [29] ([29]), this dialog represents the physical connection between artist and materials and is integral for creative cognition; IVR systems exist to enhance this dialog, not to replace it. The TASC system ([5]) effectively strikes a balance between digital resources and practical tools, where the integration of physical and virtual elements reduced cognitive load by 31% while also providing students hands-on learning experiences. For example, students learning figure drawing might start with charcoal studies to develop haptic sensitivity before moving to Tilt Brush’s volumetric sketches to enhance spatial reasoning. The alternation between physical and virtual work might offset the warning posited by [2] ([2]), in which the use of technology can overshadow the development of skills.Scaffold Embodied Experiences: The quality of embodied learning in IVR exists on a spectrum. According to [17]’s ([17]) embodiment taxonomy, which ranges from controller-based interactions (first-degree) to full-body navigation (fourth-degree), this research validates increasing learning complexity with each level of advancement. This progression suggests that creative cognition develops most effectively when IVR experiences follow the following sequence: First, students must establish sensorimotor connections using tools such as Tilt Brush to practice converting physical arm movements into virtual painted strokes, which builds basic kinesthetic proficiency needed for more complex tasks. Second, as [21] ([21]) and [42] ([42]) demonstrated, curriculum should incorporate metaphor-driven actions, like physically enacting “fluid thinking” or “breaking barriers”. These body-based abstraction techniques can strengthen creative thinking abilities without requiring advanced technical skills, proving that scaffolding need not await technical mastery. Finally, the sequence should emphasize socially embedded creation by establishing open-ended scenarios within collaborative IVR-based systems, which [25] ([25]) demonstrated can lead to individual learning through shared gestures in creative problem-solving activities.

These approaches are in line with [22]’s ([22]) call for an essential “rethinking of assessment” in embodied learning environments. They argue that existing approaches of evaluation do not take account of the nuanced ways that embodied interaction can prompt cognitive development. Rather, they emphasize that assessments should focus on the physical processes of gaining knowledge rather than on results alone. This point of view helps to inform the recommendation to develop an assessment framework based on process, which succinctly gauges students’ kinesthetic development, spatial thinking, and the ability to transfer knowledge between virtual and real settings. With the use of process-oriented assessments, educators are in a position to better analyze the multi-faceted learning that occurs as students physically interact within IVR-based systems.

#### 6.4.2. Assessment: Measuring the Immaterial

Modern assessment methods falter in the evaluation of IVR’s embodied approaches because they stick to product-based evaluation while overlooking the kinesthetic and cognitive processes essential for virtual creative education. Through their analysis of movement and gesture, [28] ([28]) demonstrated that students’ gestural adaptations (how their line quality shifts between physical and virtual media or how they recalibrate spatial decisions when tool resistance disappears) in IVR-based environments reveal more about creative development than any final digital renderings could. This suggests that the dynamic nature of learning calls for assessment approaches which match its variability.

Process Over Product: Figural drawing research confirms that meaningful evaluations might be informed by observing those transformative moments which occur when projects break the boundaries of the chosen artistic medium. Here, it is recommended that the evaluation process should utilize a comparative portfolio analysis in order to reveal the cross-modal, mixed-media-informed learning process by illuminating the iterative translations which occur during artistic development through exploration of media, methods, and processes. Such methods align with [28]’s ([28]) finding that exploratory gestures during the IVR-based activity can lead to enhanced creative problem-solving abilities.Capturing The Intangible: To move beyond superficial appraisal, educators are encouraged to use metrics which truly capture IVR’s unique educational advantages:KINESTHETIC FLUENCY MAPPING: By tracking the development from unwieldy controller manipulation to natural and confident, full-arm gestural drawing, we can gauge students’ adoption of IVR-based tools as natural extensions of their creative intent, a phenomenon known as “environmental coupling,” which serves as a key benefit of IVR-based pedagogy, according to [44] ([44]).DECISION-MAKING PATHWAY ANALYSIS: Analyzing recordings of IVR sessions can help to identify embodied ideation patterns: virtual movement around a sculpture or crouching to see their work from a different perspective demonstrates spatial reasoning that goes beyond basic navigation. This is mirrored by an analysis of students’ full-body rotations as found in [5]’s ([5]) TASC research, which shows how physical movement supports creative decision-making.MATERIAL TRANSFER BENCHMARKS: The pedagogical assessment of IVR depends on its capability to advance real-world creation practices. Does the process of digital sculpting generate more thorough clay models? Do experiments created with Tilt Brush have the potential to become bolder drawing compositions? This evaluation of the integration between virtual skills and real-world outcomes supports the essential material dialog identified by [29] ([29]) as it validates IVR’s role in creative practice.Rubric Development For Embodied Learning: The assessment approaches outlined above require formalization through specific evaluation tools that capture embodied learning’s complexity. Traditional rubrics, designed for static outputs, cannot adequately evaluate the kinesthetic intelligence developed through IVR experiences. This necessitates new frameworks that recognize physical engagement as fundamental to creative cognition:KINESTHETIC DEVELOPMENT SCALE: Tracks students’ progression from tentative controller manipulation to fluid gestural expression, building on the kinesthetic fluency mapping principle discussed earlierSPATIAL TRANSFER ASSESSMENT: Examines how virtual spatial skills translate to physical making through comparative portfolio analysis, extending the material transfer benchmarksMATERIAL DIALOG DOCUMENTATION: Requires students to articulate sensory differences through structured reflection prompts such as “How did the absence of weight affect your design decisions?”, capturing the intangible aspects that [28] ([28]) identified as crucial for creative development.

This implementation of reflective documentation aligns with [31]’s ([31]) research on portfolio-based assessment in art education, which demonstrates how structured reflection tools enable students to articulate the embodied dimensions of their creative process. Such approaches transform tacit somatic knowledge into explicit conceptual understanding, recognizing that the iterative process of acquiring knowledge through physical interaction carries equal pedagogical value to the final creative output.

However, implementing these assessment strategies demands more than updated evaluation tools. Educators themselves require preparation that extends beyond technical training to encompass the pedagogical shifts inherent in embodied virtual learning.

#### 6.4.3. Teacher Development: Beyond the Headset Manual

Many modern professional development programs fail to overcome the tool fallacy by assuming that educators who learn IVR-based tools can automatically teach with IVR. True readiness relies on what we can call pedagogical double vision: educators need the ability to guide technical proficiency while fostering student understanding of their embodied learning processes ([28]). This pedagogical sensitivity requires what [30] ([30]) terms “professional vision”, which is the ability to identify and interpret significant embodied interactions during art-making processes, enabling educators to recognize when students are developing kinesthetic understanding even when traditional assessment methods fail to capture such learning. Successful implementation of this dual focus requires two complementary skill sets rarely addressed in conventional training.

Facilitating Embodied Learning: Masterful IVR instructors bring value to their lessons by demonstrating to students the physical aspects of virtual creation. Consider a life drawing instructor helping students compare shoulder kinematics when shifting from charcoal to Tilt Brush: Where does the wrist stiffen? How does depth perception alter line intentionality? As [28] ([28]) demonstrated, teachers must recognize and nurture emergent embodied strategies and scaffold reflection through questions like “How did your virtual mark-making differ from your physical process?”. This moves beyond interface literacy to what might be termed kinesthetic pedagogy: teaching that treats the student’s body as the primary mediator of virtual experience.Mastering Interaction Typologies: Object manipulation and embodied interaction in IVR can spark unique cognitive and creative processes. For example, users who navigate through architectural models in IVR develop spatial reasoning skills by transforming theoretical concepts of perspective into interactive, immersive experiences. Additionally, activities that incorporate simulations of real materials, as with virtual clay sculpting, can not only develop haptic intuition but also encourage risk-taking and experimentation, which [11] ([11]) identifies as crucial for developing material literacy. Consequently, the role of the art teacher can evolve as they adopt strategies that embrace interactivity and embodied learning in IVR. This might involve sequencing interactions to first establish students’ spatial awareness, then moving on to material experimentation and object manipulation, ultimately leading to creative dialog. This dialog, as [29] ([29]) describe, serves as a “conversation partner” with traditional media, where each aspect informs and enriches the other in an ongoing creative exchange, which is essential to art education ([29]).

The true value of IVR for use in art and design education lies not just with the attainment of digital proficiencies but in its unique affordances in developing spatial awareness through virtually aided navigation of real or imagined spaces, its ability to ease in object manipulation and 3D visualization, and its capacity to promote creative problem-solving through embodied, interactive, and collaborative learning opportunities.

As we traverse this new terrain, and as the functionality and availability of new, more immersive and more engaging technologies develop, it is integral that teachers and institutions develop new methods to leverage these technologies in ways that revolutionize educational practices, while also maintaining didactic rigor expected in the 21st century. These principles serve not as constraints but as vital constructs ensuring that the use of IVR technology amplifies rather than replaces the material dialogs that remain the core of art education ([29]).

The pedagogical studies examined here reveal an undeniable truth: IVR is not just posed to change art education, it exposes the embodied nature of creativity in itself. As we close out this investigation, the question remains: What do these findings mean for the future of embodied learning, and what potential trade-offs might be needed?

## 7. Discussion and Future Directions

### 7.1. Contradictory Aspects of Technological Integration

As educational institutions struggle to keep up with technological advancements, the implementation of IVR-based learning paradigms represents a complex challenge for art and design educators. Although there is no question of its potential to transform arts education, teachers must consider how the technological capabilities of IVR can be applied to meet their pedagogical needs. Research into the capabilities of IVR has demonstrated its unique ability to foster creative thinking. As demonstrated in one study, students who physically enact metaphors like “breaking through” virtual barriers showed notable increases in divergent thinking, which is one of the main underpinnings of creativity ([42]). Yet these promising theoretical outcomes exist in tension with students’ lived experiences of virtual creation. When art students report that they “can’t feel the paint like real brushes” ([28]), they articulate a fundamental disconnect between IVR’s transformative potential and the material realities of artistic practice. This gap between theoretical promise and tactile experience represents the central challenge educators face when integrating virtual technologies into traditionally material-based disciplines.

### 7.2. Material Paradoxes in Virtual Creation

While one of the greatest advantages of IVR lies in its capacity to enable unlimited digital creation by freeing users from the limitations of physical materials, this is also one of its greatest weaknesses, as it often comes at the expense of tactile knowledge development. Termed “The Immateriality Paradox” ([28]), this phenomenon suggests that the same freedoms that foster artistic creativity, like weightless modeling clay or frictionless brushes, can also strip away the tactile resistance that trains artistic intuition. As demonstrated by [11]’s ([11]) study investigating hands-on instruction, blindfolded ceramics students were seen to rely on their tactile senses in order to refine their designs by determining the proper wheel-throwing pressure. Notably, this paradox has been observed during an IVR-based Tilt Brush study where users indicated that their virtual brushstrokes lacked the canvas drag which often allows for accurate, nuanced brushwork ([28]). These are not only technological issues: they reveal a fundamental gap between the tactile learning that is an integral part of the artistic process and the special affordances of IVR-based embodied learning. By highlighting this absence of materiality in IVR, educators can encourage students to consider how the material qualities of the art-making media can shape intention. With this, there is some evidence that by using hybrid educational models like the TASC system ([5]), physical materials can help to anchor virtual tasks, suggesting that this immateriality paradox can be supplanted through pedagogical strategies.

The embodied cognition framework faces particular challenges when applied to highly abstract artistic concepts. While embodied metaphors effectively support spatial reasoning and material manipulation, concepts such as “artistic voice,” “compositional balance,” or “emotional resonance” require cognitive processes that extend beyond sensorimotor grounding. Consider how students creating abstract expressionist works in IVR encounter a fundamental disconnect: although sweeping arm movements generate visual marks, the translation from physical gesture to emotional expression relies on symbolic interpretation rather than direct embodied knowledge. This limitation reveals that IVR-based embodied learning proves most effective for spatially grounded design domains (architecture, sculpture, industrial design), whereas conceptually driven artistic practices demand supplementary pedagogical approaches that bridge the gap between bodily movement and abstract meaning-making.

These limitations, both material and conceptual, point toward the need for pedagogical strategies that acknowledge IVR’s strengths while compensating for its constraints.

### 7.3. Principles for Pedagogical Implementation

The following principles for integrating embodied learning frameworks into IVR-based art education were revealed:Gestural Congruency Proves Essential: It is crucial that virtual interactions closely resemble their physical counterparts; as research on embodiment confirms, learning outcomes improve significantly when digital manipulations match real-world material behaviors ([17]). With this, we can assert that IVR-based learning is not just about mimicking physical reality, we must also preserve the kinesthetic structures that prompt learners to think with their hands.The Rhythm of Reciprocal Making: Embodied learning activities should be structured through an ongoing process, balancing material knowledge with the unrestrained freedom of virtual creation. This was found in [5]’s ([5]) TASC system, which not only prevents degradation of tactile knowledge, but actively enriches it, using IVR’s spatial affordances to reveal additional possibilities, allowing students to discover material behaviors through digital experimentation before testing them in physical form.Collaborative Embodiment as a Creative Catalyst: The use of collaborative virtual spaces can minimize feelings of isolation through what [25] ([25]) refers to as “embodied social presence”. With this, coordinated movements and co-creation of virtual objects create a tangible sense of shared experiences, which is especially important for distant learners who might otherwise miss the social setting of real art studios.

### 7.4. Emerging Solutions and Unanswered Questions

As advancements in IVR technology may result in partial solutions to the immateriality paradox, each brings with it additional conflicts. The use of haptic feedback gloves may help to bridge this gap through what [11]’s ([11]) study deemed a necessary aspect of art-based learning: a consistent dialog with the art materials. This technological adaptation also threatens to further widen the accessibility gap due to their prohibitive cost. Additionally, AR-VR hybrids, where virtual designs might be projected onto physical mediums, can sustain [29]’s ([29]) essential “material dialogue” by adding virtual flexibility. However, these innovations demand scrutiny, leading to additional questions. Can the vibration of a haptic interface be sensitive enough to approximate the nuanced sensation needed for sculpting? Or, as [44]’s ([44]) time-pressured cognition suggests, might the delay in registering a brushstroke in Tilt Brush cause a disruption in the perception–action loop, becoming a hindrance to creative freedom?

### 7.5. Critical Pathways for Future Research

These unresolved questions point to a number of essential areas for further research. First, the longitudinal studies could be implemented in order to address the issue regarding the loss of material literacy, or “haptic amnesia”, which, as suggested by [11] ([11]), comes after a long disengagement with physical materials. Can scaffolded IVR immersion aid in developing foundational skills like observational drawing, or is this forced integration a sign of “technological solutionism,” which [40] ([40]) caution. Can we successfully employ the use of IVR tools without proof of their educational value? In the absence of specific studies on the influence of IVR on fundamental artistic competencies, does its adoption place undue emphasis on novelty at the expense of meaningful skill development? Secondly, two remaining tensions described within this paper’s Limitations and Pedagogical Implementation Section may also be addressed. (1) This qualitative synthesis cannot identify causal relationships with respect to skill transfer, suggesting that further longitudinal studies may be required in order to establish whether IVR-enhanced spatial skills can enhance physical sculpting. (2) As the haptic tech progresses, the pedagogical frameworks suggested here run the risk of being obsolete before validation, as this is not just about “updating”, it is a fundamental reckoning with IVR’s cognitive trade-offs (dissociation, material disconnection) versus its creative potential for tactile thinkers. With this, future studies can also place more emphasis on embodied communication rather than technological novelty, exploring the effects of IVR-based embodiment on the sensorimotor foundations of creative cognition ([44]).

These research directions must be grounded in practical application. As we can see from Figure 3, in order to apply theories of embodied cognition to an IVR-based art and design educational environment, we can link Wilson’s theoretical frameworks with related IVR features. Here, a visual framework was generated by mapping out the findings from Phase 2 of this investigation; this details how situatedness, action-orientation, offloading, and environmental coupling can be leveraged to develop innovative pedagogical approaches, which address some of the challenges faced by learners engaged in IVR-based learning environments.

## 8. Conclusions

This theoretical synthesis aims to contribute to the ongoing literature bridging three previously disconnected domains by examining the literature relating to embodied learning, Immersive Virtual Reality, and art and design education. After examining the means by which the principles of embodied cognition present themselves within the context of art and design pedagogy, this paper has derived a theoretically grounded framework which may be useful for educators in the implementation of immersive technologies, while adhering to the fundamental embodied processes inherent in the creative domains.

Additionally, this work aims to advance the understanding of the conflicts between virtual and physical materiality inherent in IVR-enhanced design education by proposing a balanced approach which makes use of IVR’s intrinsic affordances while also pointing to its current limitations. The frameworks discussed here correlate crucial pedagogical concerns regarding sense of agency, interactive and embodied approaches, and spatial orientation, offering a new approach to understand how embodiment can inform art and design educational practices with the use of Immersive Virtual Reality.

Specifically, this paper has explored the notion that creative thinking does not just happen in the mind but through the body’s dynamic engagement with its environment. This research also reveals the conflicting nature of IVR-based embodied learning within the context of art and design education. Systems like Tilt Brush and TASC reveal how virtual embodiment can enhance spatial reasoning and creativity ([5]; [28]), but they also shed light on what we can refer to as the immateriality paradox, which is the cognitive disconnect that occurs when artistic practice is no longer grounded in physical, tactile materiality, which, according to ([11]), artists rely on in order to learn from and through various artistic mediums. The frameworks developed here aim to provide educators with a means of resolving this conflict by using IVR not as a replacement for traditional studio practice but as what can be termed a “cognitive amplifier”, which encourages creative exploration in virtual spaces, while honoring and developing students’ understanding of artistic materials and processes.

As haptic technologies advance and accessibility improves, the principles outlined here offer a theoretically grounded foundation for navigating the evolving landscape of art and design education. Success in this integration depends not merely on technological sophistication but on pedagogical wisdom in orchestrating meaningful dialogs between virtual affordances and material realities, ensuring that innovation serves rather than supplants the fundamental human processes of creative discovery.

## Figures and Tables

**Figure 1 behavsci-15-00917-f001:**
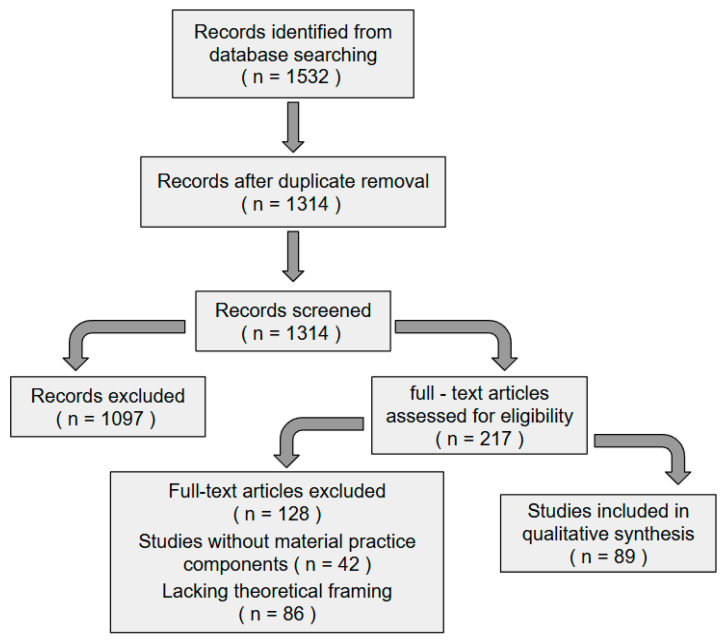
PRISMA flow diagram: The systematic review process showing identification, screening, eligibility assessment, and final inclusion of the studies.

**Figure 2 behavsci-15-00917-f002:**
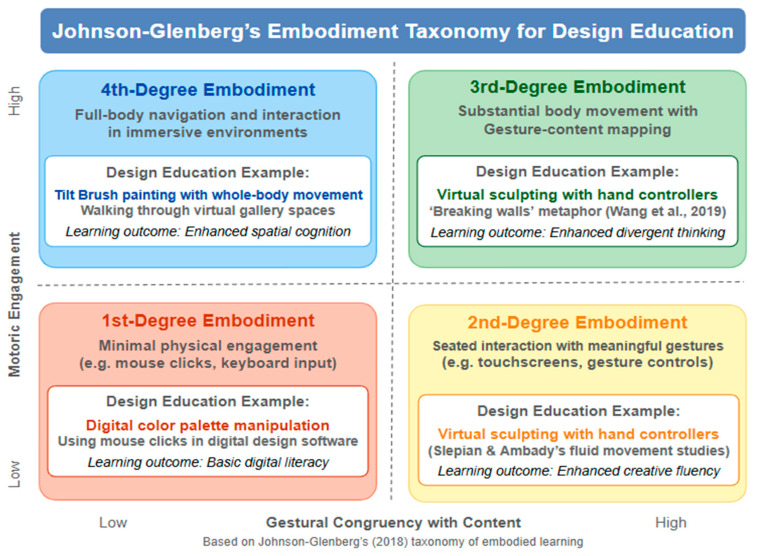
Johnson-Glenberg’s embodiment taxonomy with design education examples. The diagram illustrates the four degrees of embodiment from Johnson-Glenberg’s taxonomy with corresponding design education examples and learning outcomes ([42]; [16]).

**Figure 3 behavsci-15-00917-f003:**
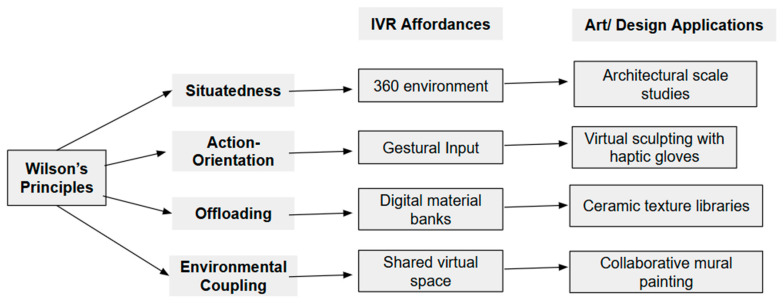
Operationalizing embodied cognition for IVR art/design pedagogy.

**Table 1 behavsci-15-00917-t001:** Case studies mapped to embodiment levels.

Study	IVR Activity	Embodiment Level	Key Pedagogical Insight
[5] ([5])	TASC spatial puzzles	third-degree	Hybrid physical–virtual interaction reduces cognitive load by 31%.
[28] ([28])	Tilt Brush painting	fourth-degree	Whole-body movement enhances creativity but risks sensory mismatch
[42] ([42])	“Breaking walls” metaphor	third-degree	Embodied metaphors increase divergent thinking

## Data Availability

This theoretical study did not generate new empirical data. All sources analyzed are cited in the references section and are publicly available through their respective journals and databases.

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
