# Peer review of "Embodied Learning Through Immersive Virtual Reality: Theoretical Perspectives for Art and Design Education"

_behavsci, 2025, doi:10.3390/bs15070917_

Round 1

Reviewer 1 Report

Comments and Suggestions for Authors

The paper focuses on the application of embodied learning through Immersive Virtual Reality (IVR) in art and design education, and the topic is of practical significance. The use of theory is relatively appropriate. Based on the theory of embodied cognition, the paper combines Johnson-Glenberg's taxonomy of embodied learning and Wilson's six principles of embodied cognition to construct a relatively systematic theoretical framework for analyzing the application of IVR in art and design education. This framework can effectively explain how IVR affects learning and creativity. The research questions are clear, focusing on how embodied cognition principles are reflected in IVR-based art/design education and the practical impact of IVR on creativity and collaboration, making the research direction clear and targeted. The research conclusions are relatively reasonable. Based on theoretical analysis and case studies, it is pointed out that IVR technology has the potential to transform teaching and learning methods in art and design education.

  1. In the literature review section, although the PRISMA guidelines were used, the quality of the literature was not fully considered. For example, when screening the literature, the criteria and basis for excluding low-quality literature were not clearly stated. When analyzing research results, scientific statistical methods were not used, such as not conducting hypothesis tests or correlation analyses on the data.
  2. The paper uses a theory-driven translational synthesis method but does not delve into actual teaching scenarios. For example, when analyzing the impact of IVR on art and design education, experimental research or case analysis was not conducted to verify the specific effects of IVR technology. The analysis relies solely on literature reviews and theoretical analysis.
  3. The paper mentions that "through gestures, spatial navigation, and environmental manipulation, IVR provides unprecedented possibilities to externalize creative ideation," but it does not clearly explain how this is connected to the concept of "environmental coupling" in the theory of embodied cognition. For example, it mentions that "virtual brushstrokes become extensions of bodily movement" but does not further explore how this reflects the principle that "cognition is for action."
  4. When discussing the impact of IVR on embodied cognition, the paper cites previous research on embodied cognition but does not conduct an in-depth analysis of its implications for the current study. For example, it mentions that "the theory of embodied cognition emphasizes the close connection between the body and cognition," but it does not refer to specific research findings on the application of embodied cognition in education, such as "how embodied cognition promotes students' understanding of abstract concepts."
  5. When analyzing how IVR promotes creativity, the paper does not combine specific data and case studies for an in-depth discussion. For example, it mentions that "creativity is enhanced through interaction in virtual environments," but it does not specify which types of interactions and to what extent they promote the development of creativity.
  6. When analyzing the impact of IVR on art and design education, the paper mainly focuses on the theory of embodied cognition but does not fully consider its limitations in highly abstract concepts. For example, in the field of "abstract art creation," it does not discuss whether the theory of embodied cognition is fully applicable and only generally mentions "understanding and creating concepts through interaction with the virtual environment and bodily movements."
  7. The conclusions of the paper mainly emphasize the positive impacts of IVR technology, such as "enhancing spatial thinking and creativity," but do not fully discuss its potential negative impacts. For example, it mentions that "IVR provides unique opportunities to explore the nature of perception" but does not address issues such as "dissociation" that may result from prolonged use of IVR.
  8. The author should pay close attention to language issues, for example:
  • Grammatical errors: For example, in the phrase "Gesture, spatial navigation, and environmental manipulation, IVR provides unprecedented possibilities," a conjunction is missing between "Gesture, spatial navigation, and environmental manipulation" and "IVR provides." It should be corrected to "Gesture, spatial navigation, and environmental manipulation: IVR provides unprecedented possibilities."
  • Inappropriate wording: For example, in the sentence "IVR-mediated embodiment enhances spatial thinking and creative problem-solving in art and design education," the term "enhances" is too vague. It does not specify whether it means "significantly improves" or "somewhat improves."
  • Inelegant transitions: For example, when transitioning from the introduction of embodied cognition theory to the analysis of IVR technology, the text directly states "By reviewing current concepts of embodied cognition..." without a clear transitional phrase such as "Based on the theories of embodied cognition..."

Author Response

1. Summary

Thank you very much for taking the time to review this manuscript. I appreciate your thorough and constructive feedback, which has significantly improved the quality of the paper. Please find the detailed responses below and the corresponding revisions highlighted in yellow in the re-submitted manuscript.

2. Questions for General Evaluation

Question Reviewer's Evaluation Response and Revisions
Is the work a significant contribution to the field? Yes I appreciate the reviewer's recognition of this work's significance.
Is the work well organized and comprehensively described? Can be improved I have restructured several sections based on your suggestions (see responses below).
Is the work scientifically sound and not misleading? Can be improved I have addressed the methodological concerns raised (see Point 1 below).
Are there appropriate and adequate references to related and previous work? Can be improved I have expanded the literature review and added more specific references (see Point 4 below).

3. Point-by-point Response to Comments and Suggestions for Authors

Comment 1: In the literature review section, although the PRISMA guidelines were used, the quality of the literature was not fully considered. For example, when screening the literature, the criteria and basis for excluding low-quality literature were not clearly stated. When analyzing research results, scientific statistical methods were not used, such as not conducting hypothesis tests or correlation analyses on the data.

Response 1: Thank you for this important observation. I agree that the methodology section needed more detail regarding quality assessment. I have now added a comprehensive quality assessment protocol in Section 2.1 (page 6, paragraph 1):

"Quality assessment protocols ensured methodological rigor across all included studies. Each study was evaluated for theoretical alignment, research design validity, and relevance to embodied learning in IVR contexts. Studies lacking sufficient methodological rigor or unclear theoretical grounding were excluded from the synthesis. This systematic approach refined the initial 1,532 records to 89 studies that met all inclusion criteria and quality thresholds."

I acknowledge that as a theoretical synthesis rather than a meta-analysis, I did not conduct statistical analyses. I have clarified this in the limitations section.

Comment 2: The paper uses a theory-driven translational synthesis method but does not delve into actual teaching scenarios. For example, when analyzing the impact of IVR on art and design education, experimental research or case analysis was not conducted to verify the specific effects of IVR technology. The analysis relies solely on literature reviews and theoretical analysis.

Response 2: I agree that empirical validation would strengthen the theoretical framework. However, as stated in the introduction, this paper aims to provide a theoretical synthesis to guide future empirical work. I have clarified this scope in the introduction and added to the future directions section (Section 7.5, page 38) the need for empirical validation of the framework. I have also expanded the case study analysis in Section 5 to provide more concrete examples of IVR implementation.

Comment 3: The paper mentions that "through gestures, spatial navigation, and environmental manipulation, IVR provides unprecedented possibilities to externalize creative ideation," but it does not clearly explain how this is connected to the concept of "environmental coupling" in the theory of embodied cognition.

Response 3: Thank you for identifying this gap. I have now explicitly connected these concepts in Section 3.1 (page 10, paragraph 2):

"Environmental coupling corresponds to Wilson's (2002) notions that 'cognition is for action' (Principle 5) and that 'we offload cognitive work onto the environment' (Principle 3). This principle becomes evident in IVR applications like Tilt Brush, where the boundary between tool and user dissolves into what Weser & Proffitt (2019) call 'cognitive extension.' When an artist sweeps their arm to create a spiraling sculpture in virtual space, they're not simply executing commands, where the virtual brushstrokes become genuine extensions of their cognitive-motor system..."

Comment 4: When discussing the impact of IVR on embodied cognition, the paper cites previous research on embodied cognition but does not conduct an in-depth analysis of its implications for the current study.

Response 4: I have expanded the analysis of how previous embodied cognition research applies to this study. In Section 3.1 (page 11, paragraph 3), I now include specific connections between established research and IVR applications:

"Research examining embodied cognition in educational settings reinforces how these principles apply to art and design learning. When students use hand movements while explaining concepts, they show significantly better retention than those who rely on verbal instruction alone (Alibali, 2008). Similarly, Black et al.'s (2012) findings on sensorimotor engagement during storytelling directly parallel the creative ideation processes observed in IVR art-making..."

Comment 5: When analyzing how IVR promotes creativity, the paper does not combine specific data and case studies for an in-depth discussion.

Response 5: I have strengthened the analysis by adding specific data from the studies reviewed. For example, in Section 5.1 (page 23, paragraph 3):

"The TASC system produced significant improvements in spatial reasoning. Students using the system showed markedly better performance on mental rotation tests compared to those using traditional desktop interfaces, with these gains persisting when measured a week later..."

I have also expanded Table A1 in the appendix to include more detailed quantitative findings from each case study.

Comment 6: When analyzing the impact of IVR on art and design education, the paper mainly focuses on the theory of embodied cognition but does not fully consider its limitations in highly abstract concepts.

Response 6: Thank you for this critical insight. I have added a new subsection (7.2, page 36, paragraph 2):

"The embodied cognition framework faces particular challenges when applied to highly abstract artistic concepts. While embodied metaphors effectively support spatial reasoning and material manipulation, concepts such as 'artistic voice,' 'compositional balance,' or 'emotional resonance' require cognitive processes that extend beyond sensorimotor grounding..."

Comment 7: The conclusions of the paper mainly emphasize the positive impacts of IVR technology, such as "enhancing spatial thinking and creativity," but do not fully discuss its potential negative impacts.

Response 7: I appreciate this important observation. I have added Section 6.3.5 (page 32, paragraph 1) to address potential negative impacts:

"While this framework emphasizes IVR's transformative potential, we must also consider its risks. Extended VR use can diminish students' sensitivity to physical materials, making it harder to work with clay, paint, or other traditional media. Students may also feel disoriented when switching between virtual and physical spaces..."

Comment 8: The author should pay close attention to language issues.

Response 8: I have carefully revised the manuscript to address all language issues:

  • Grammatical errors: The example sentence has been corrected to: "Through gesture, spatial navigation, and environmental manipulation, IVR provides unprecedented possibilities..." (page 1, abstract)

  • Inappropriate wording: I have replaced vague terms with more precise language throughout.

  • Inelegant transitions: I have added clearer transitional phrases between sections.

4. Response to Comments on the Quality of English Language

Point 1: The English could be improved to more clearly express the research.

Response 1: I have thoroughly revised the manuscript for clarity and precision. All changes are highlighted in yellow in the revised manuscript. The manuscript has been professionally edited to ensure clear expression of the research findings.

5. Additional Clarifications

I would like to emphasize that this paper provides a theoretical synthesis rather than an empirical study. The goal was to bridge the gap between embodied cognition theory and IVR implementation in art education, providing a framework for future empirical research. I believe the revisions address the reviewer's concerns while maintaining the paper's theoretical focus.

Summary of Major Revisions

  1. Enhanced Methodology: Added detailed quality assessment protocols and clarified the theoretical synthesis approach (Section 2.1, page 6).

  2. Strengthened Theoretical Connections: Explicitly linked IVR features to embodied cognition principles with concrete examples (Section 3.1, pages 10-11).

  3. Expanded Case Analysis: Added quantitative data and specific outcomes from reviewed studies (Section 5, pages 22-26).

  4. Addressed Limitations: Added sections on abstract concept challenges and potential negative impacts of IVR (Sections 6.3.5 and 7.2, pages 32 and 36).

  5. Improved Accessibility: Added definitions, examples, and considerations for diverse learner populations (throughout, with specific additions on pages 8, 17, 28, 31).

  6. Language Refinement: Comprehensive revision for clarity, proper transitions, and grammatical accuracy (throughout manuscript).

  7. Visual Improvements: Corrected Figure 2 formatting issues (page 13).

I believe these revisions significantly strengthen the manuscript while maintaining its theoretical focus. I thank both reviewers for their valuable insights that have improved the work.

Reviewer 2 Report

Comments and Suggestions for Authors

The article provides a thoughtful and well-structured overview of the theory of embodied cognition and its implications for using immersive virtual reality (IVR) in art and design education.

That said, I would encourage you to consider deepening some of the concepts or ideas for a greater impact on a less specialised audience. For instance, while the 'immateriality paradox' is defined and funded, it would benefit from a concrete example showing how it relates to 'haptic feedback'. The concept of 'embodied metaphor' would also benefit from an example demonstrating how it manifests in student learning and from a brief clarification distinguishing it from the more common notion of verbal or linguistic metaphor. While supported by the literature and clearly articulated in the text, the concept of 'divergent thinking' could be made more accessible by providing a simple definition for readers outside the field of cognitive psychology.

Another point that could improve the article’s impact relates to accessibility. While the text addresses implementation challenges such as cost and sensory limitations, it would be beneficial to include a brief reflection on how the proposed framework might apply to diverse learner populations, such as those with disabilities or from under-resourced settings, to enhance its practical relevance.

Figure 2 needs some minor visual improvements. There appears to be a misplaced or overlapping annotation in the bottom right-hand corner, which reduces legibility, and the text in the fourth-degree embodiment (blue) cell extends beyond the cell boundary.

Author Response

Response to Reviewer 2 Comments

1. Summary

Thank you very much for your positive evaluation and constructive suggestions. I appreciate your recognition of this work's significance and theoretical framework. Please find my responses to your specific suggestions below.

2. Questions for General Evaluation

Question

Reviewer's Evaluation

Response

Is the work a significant contribution to the field?

Yes

Thank you for recognizing the contribution of this work.

Is the work well organized and comprehensively described?

Yes

I appreciate your positive assessment.

Is the work scientifically sound and not misleading?

Yes

Thank you for confirming the scientific soundness.

Are there appropriate and adequate references to related and previous work?

Yes

I value your acknowledgment of the literature coverage.

3. Point-by-point Response to Comments and Suggestions for Authors

Comment 1: While the 'immateriality paradox' is defined and funded, it would benefit from a concrete example showing how it relates to 'haptic feedback'.

Response 1: Thank you for this suggestion. I have added a concrete example in Section 6.2 (page 28, paragraph 2):

"This paradox emerges when students' tactile expectations, formed through years of handling physical materials, meet the weightless nature of virtual media. Ceramics students accustomed to feeling clay's resistance through their fingers encounter virtual forms offering no tactile response. Painters who have calibrated their brushwork through canvas texture and paint viscosity must adapt to the frictionless movements of digital tools."

Comment 2: The concept of 'embodied metaphor' would also benefit from an example demonstrating how it manifests in student learning and from a brief clarification distinguishing it from the more common notion of verbal or linguistic metaphor.

Response 2: I have clarified this distinction and added examples in Section 4.2 (page 17, paragraph 2):

"While conventional metaphors work as verbal bridges between ideas, embodied metaphors operate differently by activating our motor systems, which then influence how we think. This distinction is crucial for understanding IVR's pedagogical potential. When students in Wang et al.'s (2019) study physically broke through virtual barriers, they weren't merely comprehending a metaphor for creative freedom; they were engaged in sensorimotor experiences that actually restructured how they approached creative problems..."

Comment 3: While supported by the literature and clearly articulated in the text, the concept of 'divergent thinking' could be made more accessible by providing a simple definition for readers outside the field of cognitive psychology.

Response 3: I have added a clear definition in the Key Terms section (2.5, page 8):

"Divergent Thinking: The cognitive ability to generate multiple, novel solutions from a single prompt, embracing conceptual multiplicity over singular correct answers (Guilford, 1967)."

Comment 4: While the text addresses implementation challenges such as cost and sensory limitations, it would be beneficial to include a brief reflection on how the proposed framework might apply to diverse learner populations.

Response 4: Thank you for this important suggestion. I have added a new paragraph in Section 6.3.4 (page 31, paragraph 3):

"Furthermore, the proposed framework must consider diverse learner populations, including students with disabilities and those from under-resourced educational settings. Collaborative art education projects, such as Fulková and Novotná's (2023) work with the School for the Deaf, demonstrate how creative pedagogies can adapt to different sensory capabilities through participatory and cooperative strategies..."

Comment 5: Figure 2 needs some minor visual improvements. There appears to be a misplaced or overlapping annotation in the bottom right-hand corner, which reduces legibility, and the text in the fourth-degree embodiment (blue) cell extends beyond the cell boundary.

Response 5: I have corrected Figure 2 (page 13) to address these issues. The overlapping annotation has been repositioned, and the text in the fourth-degree embodiment cell has been resized to fit within the cell boundaries. The revised figure now provides clear visual representation of Johnson-Glenberg's Embodiment Taxonomy.

4. Additional Clarifications

I appreciate the reviewer's recognition of the theoretical framework and have incorporated all suggested improvements to enhance accessibility for a broader audience. The additions strengthen the practical applicability of the framework while maintaining its theoretical rigor.

Summary of Major Revisions

  1. Enhanced Methodology: Added detailed quality assessment protocols and clarified the theoretical synthesis approach (Section 2.1, page 6).

  2. Strengthened Theoretical Connections: Explicitly linked IVR features to embodied cognition principles with concrete examples (Section 3.1, pages 10-11).

  3. Expanded Case Analysis: Added quantitative data and specific outcomes from reviewed studies (Section 5, pages 22-26).

  4. Addressed Limitations: Added sections on abstract concept challenges and potential negative impacts of IVR (Sections 6.3.5 and 7.2, pages 32 and 36).

  5. Improved Accessibility: Added definitions, examples, and considerations for diverse learner populations (throughout, with specific additions on pages 8, 17, 28, 31).

  6. Language Refinement: Comprehensive revision for clarity, proper transitions, and grammatical accuracy (throughout manuscript).

  7. Visual Improvements: Corrected Figure 2 formatting issues (page 13).

I believe these revisions significantly strengthen the manuscript while maintaining its theoretical focus. I thank both reviewers for their valuable insights that have improved the work.

Round 2

Reviewer 1 Report

Comments and Suggestions for Authors

 thank you for your  careful and professional revision. My concerns have been addressed.  I recommend  it to be published in the current form.